# A Scoping Review on Air Quality Monitoring, Policy and Health in West African Cities

**DOI:** 10.3390/ijerph17239151

**Published:** 2020-12-07

**Authors:** Celia Mir Alvarez, Renaud Hourcade, Bertrand Lefebvre, Eva Pilot

**Affiliations:** 1Department of Health, Ethics and Society, Faculty of Health, Medicine and Life Sciences (FHML), Care and Public Health Research Institute (CAPHRI), Maastricht University, 6229 ER Maastricht, The Netherlands; cmiral97@gmail.com; 2University Rennes, EHESP, CNRS, ARENES–UMR 6051, F-35000 Rennes, France; renaud.hourcade@ehesp.fr (R.H.); bertrand.lefebvre@ehesp.fr (B.L.)

**Keywords:** urban air pollution, West Africa, ECOWAS, public health, air quality monitoring, air quality standards

## Abstract

Ambient air pollution is a global health threat that causes severe mortality and morbidity from respiratory, cardiovascular, and other diseases. Its impact is especially concerning in cities; as the urban population increases, especially in low- and middle-income countries, large populations risk suffering from these health effects. The Economic Community of West African States (ECOWAS) comprises 15 West African countries, in which many cities are currently experiencing fast growth and industrialization. However, government-led initiatives in air quality monitoring are scarce in ECOWAS countries, which makes it difficult to effectively control and regulate air quality and subsequent health issues. A scoping study was performed following the Arksey and O’Malley methodological framework in order to assess the precise status of air quality monitoring, related policy, and legislation in this region. Scientific databases and gray literature searches were conducted, and the results were contrasted through expert consultations. It was found that only two ECOWAS countries monitor air quality, and most countries have insufficient legislation in place. Public health surveillance data in relation to air quality data is largely unavailable. In order to address this, improved air quality surveillance, stricter and better-enforced regulations, regional cooperation, and further research are strongly suggested for ECOWAS.

## 1. Introduction

Air pollution has become an increasingly alarming problem in many cities across the globe. The World Health Organization (WHO) stated that in 2016 “more than 80% of people living in urban areas that monitor air pollution are exposed to air quality levels that exceed [WHO] limits” [1]. The health-related consequences of this have been well established over recent years: air pollution, especially particulate matter—referred to as PM_2.5_ and PM_10_ when the particle diameters measure less than 2.5 or 10 micrometers, respectively—can affect both respiratory and cardiovascular health through conditions, such as chronic obstructive pulmonary disease, myocardial infarction, stroke, and cancer [2,3,4]. In addition, new links are being discovered between harmful PM pollution and diabetes, decreased neurological function in children, and increased risk of mortality in Covid-19 patients [4,5]. Gaseous air pollutants that are expelled through anthropogenic activities, namely nitrogen oxides (NO_x_), sulfur dioxide (SO_2_), Black Carbon (CO), and ozone (O_3_), have also been shown to increase cardiovascular and respiratory mortality and morbidity [6,7]; these air pollutants further harm human health by affecting the environment through phenomena, such as acid rain [8]. As the urban population increases, especially in low- and middle-income countries (LMICs), an increasing amount of people are becoming susceptible to suffering from these health effects [8]. A study that was carried out in Senegal, for example, identified the links between poor air quality and conditions, like asthma and bronchitis, especially in urban regions [9]. Air pollution also leads to millions of preventable deaths each year—4.2 million in 2016—90% of which are concentrated in LMICs [2,10].

Since the harmful effects of air pollution became evident, high-income countries, the WHO, and the United Nation’s (UN) Sustainable Development Goals have adopted or endorsed clean air guidelines and strategies [4,11]. This has helped many parts of the world today to develop effective policy means and data to control, and even reduce, harmful emissions in cities and, consequently, limit health burdens [2]. This trend includes countries that have known harsh air pollution situations until recently, such as China [12,13]. However, in other regional contexts, for varying reasons, governments continue to struggle with increasing levels of urban air pollution and underperforming air quality surveillance. A clear example is the Economic Community of West African States (ECOWAS), which comprises fifteen West African nations (see map in Figure 1). These countries are subject to unique meteorological and geographical phenomena that greatly influence air quality, such as highly seasonal weather (a rainy season between November and March, and dry season the rest of the year) and close proximity to the Sahara Desert. Especially during the dry season, most ECOWAS cities receive large amounts of Saharan dust, while others suffer from smoke from Central African forest and agricultural fires, the effects of which have not yet been widely studied for this part of the world [14,15]. These long-range pollution sources are added to the rapid population growth, industrialization, urbanization, and motorization that have recently taken ECOWAS cities by storm (see Table 1), greatly increasing anthropogenic emissions due to traffic and other activities that are related to urbanization such as waste burning [16]. Therefore, air pollution levels in ECOWAS cities are quickly rising to dangerous levels: Nigerian cities, like Onitsha and Kaduna, are now among the most polluted in the world, with PM_10_ levels that are 30 and 21 times over WHO’s limit, respectively [17]. To make matters worse, all ECOWAS nations are classified as low- or lower-middle income countries [18], which means that their public health system is fragile and their population is more susceptible to poverty-related diseases, such as tuberculosis [19]. Studies have shown that long-term exposure to ambient air pollution increases the risk of developing active and even drug-resistant tuberculosis infections, putting the inhabitants of ECOWAS cities further at risk [20,21,22].

In general, the availability of air quality data from government sources is an issue for ECOWAS countries. This contributes to the problem of urban air pollution receiving low public attention and being placed low on governmental agendas. Consequently, public bodies frequently lack legal or political incentives, funds, standards, and policy guidelines to help them keep air quality within healthy ranges [26]. In the countries where legislations have been passed, enforcement is often weak, as well as coordination between relevant government sectors. For example, Toure et al. [9] showed how coordinating data from the Ministry of the Environment and the Ministry of Health would allow for better understanding the extent of the influence of air quality on respiratory conditions in Senegal, yet it is unknown whether or not such coordination occurs at the government level for Senegal or for other ECOWAS countries. Insufficient air quality data means that many national and local governments only have partial knowledge of emissions sources, concentrations, and evolution trends. It also means the effectiveness of predictive models that are used by some governments may be limited, as they require ground-level measurements to be accurate [27]. This deficient knowledge regarding the main sources of pollution makes the creation of emissions inventories, which are a necessary input for these models, very challenging [28]. It also hampers the selection of target values and the setting of priorities that are adapted to the local context. In sum, without such monitoring tools, addressing the health impacts of urban air pollution poses immense difficulties for policy makers.

In terms of official air quality monitoring and related health knowledge, ECOWAS countries lag behind other African ones, such as South Africa, for which studies using air quality monitoring data in order to assess health risks exist [29]. Offor et al. [30], for example, gathered a list of all air quality studies that were carried out for Nigeria between 1985 and 2015, all of which acquired their data independently. Another study in Senegal deployed low-cost sensors of their own to assess air quality in Dakar [31]. ECOWAS countries have also been the subject of several large-scale initiatives aimed at characterizing and understanding air quality in the region, such as the Aerosol Robotic Network (AERONET), the African Monsoon Multidisciplinary Analysis (AMMA), or the Pollution des Capitales Africaines program; this project identified the NO_2_ and Black Carbon levels—with the latter being a strong indicator for urban air pollution—exceeding WHO limits in Dakar (Senegal), Bamako (Mali), Ouagadougou (Burkina Faso), and Cotonou (Benin) between 2007 and 2010 [32,33]. The Dynamics-Aerosol-Chemistry-Cloud interactions in West Africa (DACCIWA) project advocated for the combination of air quality and health data by using hospital admissions from certain respiratory or cardiovascular diseases as an indicator of long-term exposure to PM_2.5_ [27]. These projects provided much-needed indications of the situation regarding the air quality of these countries, but their means were limited, and their research inevitably time-bound. As such initiatives end, ECOWAS’s main cities continue to grow and industrialize at a rapid pace, while the natural pollution sources of the area continue to exert their effects on the population, but without scientific measurements that could provide a vital understanding of the evolution of these phenomena.

Controlling levels of urban air pollution—and, therefore, controlling health-related consequences—starts with sound research. To date, no comprehensive literature review of available knowledge on this subject has been performed, as the few existing reviews on air quality and health in the Sub-Saharan African region (none were specific to West Africa) have been centered around indoor air pollution [34] or on specific population, such as children [35], and have not assessed the status of air quality monitoring or policy. Moreover, gray literature and expert knowledge are essential for comprehensively gathering existing information, as scientific literature for this topic and this region is limited. Against this backdrop, the present study offers a recension of existing knowledge regarding air quality monitoring networks in ECOWAS cities, evaluates current trends in the development of such networks, and reviews related policy initiatives and health effects for all fifteen countries. Such a scoping study appears to be a necessary step in better understanding where this region currently stands, identify information gaps, and highlight future research and cooperation strategies.

## 2. Materials and Methods

The present research was conducted in the form of a scoping study. Levac et al. [36] state that “researchers can undertake a scoping study to examine the extent, range and nature of research activity”, which aligns with the objectives of this study. The Arksey and O’Malley methodological framework [37] was followed, when considering additional suggestions to this framework that were provided by Levac et al. [36]. The framework was complemented with the Preferred Reporting Items for Systematic reviews and Meta-Analyses extension for Scoping Reviews (PRISMA-ScR) checklist, which was developed to ensure “methodological and reporting quality” [38]. The PRISMA-ScR checklist can be consulted in Appendix B. The Arksey and O’Malley framework is divided into the six steps that are summarized below.

### 2.1. Identifying the Research Question

A broad, main research question was formulated: *What is the present state of air quality monitoring, policy, and health effects in ECOWAS cities?* More specific sub-questions were also formulated in order to guide the literature search:What are the main contributors to ambient air pollution in ECOWAS cities?Which ECOWAS cities currently have air quality monitoring sites in place?What standards or regulations exist regarding air quality in ECOWAS cities?Are links between air quality and health known and considered in air pollution policy?Are there ECOWAS cities that can be regarded as an example for others to follow?

### 2.2. Identifying Relevant Studies

The research questions were translated into a list of search terms that aimed to encompass all vocabulary related to polluting activity, air quality monitoring, West African cities, and health. Appendix A and Table A1 depict the full search strategy. Searches were conducted in PubMed, the Web of Science, Scopus, and the French database BibCNRS. For this database, two separate searches were conducted, one using the search terms in English and one with the terms translated into French. Searches were conducted throughout the month of June 2020.

Additional to literature databases, government websites of all 15 ECOWAS countries were searched in order to identify relevant policies or documents. The WHO, World Bank, ECOWAS, United Nations Environment Programme (UNEP), and African Development Bank websites were also searched in order to gather relevant gray literature. Websites of organizations and coalitions working on air pollution were also scanned: Abidjan (Ivory Coast) is a member city of the Global Urban Air Pollution Observatory (GUAPO); and the Climate and Clean Air Coalition (CCAC) is partnered with several ECOWAS countries in order to support air pollution-related initiatives. Given the low amount of scientific literature available for this part of the world, searching these gray literature sources was essential to ensure that all relevant information was gathered. Finally, reference lists of included studies were also examined. The results were saved and inclusion and exclusion criteria were determined *post hoc*, as contemplated within the scoping review framework [37].

### 2.3. Study Selection

Scientific literature, reports, guidelines, websites, and other similar documents in English or French were eligible for inclusion. Literature was excluded if it was published before 1990, if no ECOWAS city (population of 500.000 inhabitants or more, or a capital city) was included, if indoor air pollution (rather than outdoor or ambient air pollution) was the main topic, or if other forms of pollution were more predominant topics than air pollution. Studies were included for review if they focused on an ECOWAS city and provided information regarding important polluting activities or official monitoring sites, if they mentioned government policies, standards, or regulations concerning air quality, or if they linked air quality monitoring data to effects on public health. As long as a study included a section providing any of this information, it was included, even if the study’s main goals did not match those of this scoping review. The map in Figure 1 presents the included cities.

### 2.4. Charting the Data

Articles were classified in a spreadsheet according to the information that they provided on air pollution in ECOWAS cities. Figure 2 summarizes the number of sources that are associated to each category. If a study touched upon more than one category, the most prominent topic addressed by it was selected for categorization. Subsequently, studies were reviewed, and relevant data were extracted. Appendix B
Table A3 gathers all included studies, including the category that they were assigned to.

### 2.5. Collating, Reporting, and Summarizing the Results

The results were organized according to the categories that are described in Figure 2. All air quality monitoring sites and polluting sources were identified and summarized in the Results section, as well as standards and expert insights found to be related to air pollution in ECOWAS countries. Relationships to health that were identified were highlighted within each results subsection.

### 2.6. Consultation

Step six of the scoping review methodological framework allows for optional consultation to “provide insights beyond those in the literature” [36]. The low amount of written information available on the research topic highlights the need for expert interviews to fill in gaps and provide precise knowledge that is backed by experience. Interviews were held online, and they were transcribed while using the otter.ai software. One participant submitted written responses.

Experts were identified during the literature review and gray literature search: the participants were professionals with proven experience researching or working in, one or more ECOWAS countries. Experts were questioned on their knowledge of monitoring stations, policy, polluting activity, or health implications of urban air pollution in their ECOWAS country of expertise. In order to address their findings, identification (ID) numbers were used to name each participant and they are referenced in the Results and Discussion sections based on Table 2, below. Interviews were held throughout July 2020. Interviews were transcribed, anonymized, and later analyzed in order to incorporate findings into the results.

## 3. Results

A total of 745 articles were uncovered after combining results from all databases and removing duplicates. Gray literature searches uncovered a total of 30 eligible documents and websites. Titles were screened, after which a large number of studies could be removed if the title mentioned certain exclusion criteria, such as incorrect geographical location or a focus on indoor air pollution or rural areas. After this step, 246 studies remained for abstract screening. Of these, the full texts of 38 were scrutinized. Seven studies were excluded: the full text was unavailable for one, four did not contain information specific enough pertaining to monitoring sites or regulations, and two contained general information that was better described by another source. Thirty-one were finally included for review. These numbers are summarized in the flowchart found in Figure A1 in Appendix B, and the findings are summarized in Figure 2 and Table 3, below.

### 3.1. Polluting Sources

The extent of polluting activities in the ECOWAS region is, on a general scale, known and it has been studied and documented. It is more difficult to identify particularities at the country level, and even more so to determine city-specific air pollution issues within the available literature. Natural air pollution sources cannot be forgotten either: ID4 states how in Zaria (Nigeria), Saharan dust greatly contributes to PM levels in the city.

A growing vehicle fleet is an issue for all rapidly urbanizing ECOWAS cities, especially for those of a larger size. The percentage of ambient air pollution attributable to this source is extremely important in some cases: in Ghana, Schwela [26] finds that vehicles are responsible for 80% of the nation’s ambient air pollution. The reason for these extreme vehicle emissions often relates to the average age of the vehicles being used in these countries. In Accra, 96% of vehicle emissions are produced by vehicles that were manufactured prior to 1996 [39], and in Burkina Faso the average vehicle is 14 years old. Two-wheeled vehicles are becoming increasingly present in this country, as well as in Benin, Mali, and Nigeria. In Lagos, 43% of ambient air pollution between 2003 and 2007 was due to vehicles, and the city hosted 40% of the nation’s new vehicle registrations [40]. However, Amegah and Agyei-Mensah [41] evaluated that the Bus Rapid Transit system implemented in Lagos in 2008 did lead to an improvement in air quality. Doubtful or irregular quality of fuel in West Africa further deteriorates the situation. The unpaved roads that generate dust as vehicles pass over them also play a relevant role [26,42].

Increasing industrial activity is an additional issue for most major ECOWAS cities. In Conakry (Guinea), power plants and cement or plastic factories are widely present. In Bamako (Mali), industrialization is also gaining importance as a pollution source. Mining and cement industries are very present in Togo. 90% of Senegal’s industries are located in, or near, the city of Dakar, with the emissions exceeding the nation’s own air quality standards [26,31]. In Nigeria, large cities often host a range of industrial sectors that threaten air quality. For instance, Onitsha and Enugu are home to automobile, plastic, and paint industries that are located in close proximity to residential areas [43]. In the Warri metropolis, petrochemical, oil, and gas flaring activities also occur very close to densely populated city areas [44]. Lagos, which is ECOWAS’s largest city, hosts up to 60% of Nigeria’s non-oil industries [40]. The oil sector has been extremely harsh on the air quality of Port Harcourt, where excessive gas flaring in 2017 led to what Nigeria’s Federal Ministry of the Environment declared as an “emergency situation” [17]. Gas flaring and other oil-related activities—often times illegal or unregulated—are responsible for extremely high levels of ambient air pollution in the entire Niger Delta [45].

Other important sources of air pollution in ECOWAS cities are related to meeting the energy demands of rapidly growing populations, which leads to increased fossil fuel burning. In Nigeria, the energy distribution network is so unreliable that most homes, offices, and industries have individual diesel-powered generators to cover their energy needs [ID5]. Uncontrolled open burning and waste management, in general, are also important contributors to ambient air pollution in cities, especially in the Ivory Coast, Nigeria, and Ghana [26,39,45], [ID3–ID5]. In Kano (Nigeria), desertification through burning and other forms of deforestation has increased dust aerosol pollutants in the city [42].

Lastly, a pollution source that continues to be of extreme importance in ECOWAS cities is household air pollution. Although often studied as a separate phenomenon, especially when it comes to its health effects, indoor air-polluting activities also contribute a great deal to the overall urban ambient air quality [ID3]. This effect is strongest in lower-income neighborhoods of ECOWAS cities, such as Accra (Ghana), where indoor cooking from wood or coal and heating activities that produce emissions are a common feature of domestic life [46,47], [ID2].

Little or no information on main polluting activities was identified for Cabo Verde, The Gambia, Guinea-Bissau, Liberia, Niger, and Sierra Leone. Nevertheless, general ECOWAS-level literature and cross-border influences allow for the assumption that natural pollution sources, along with the above-mentioned pollution sources that are related to population growth and urbanization, may also be relevant for these countries.

Mentions to official data on adverse effects on health that are specifically related to polluting activities in ECOWAS cities are scant. However, it seems that most cities have experienced an increase in conditions, such as asthma and bronchitis, over recent years [26]. In Ghana, a study that was conducted by the Ghana Environmental Protection Agency (EPA) and the Ghana Health Service concluded that children living in high-traffic urban settings had more visits to the hospital with upper respiratory tract infections [26]. The effects of air pollution on children are also especially strong in Nigeria, although research on the matter is lacking [40]. Eye irritations are also common in areas with substantial traffic [26,44]. Proximity to industrial sites increases not only exposure to emissions, but also other toxic substances [26]. The long-term health effects of urban air pollution exposure in ECOWAS seem to be undocumented.

### 3.2. Monitoriong Stations and Data Availability

Accra (Ghana) is one ECOWAS city that has recently made important progress regarding air quality monitoring. The Ghana Environmental Protection Agency, in collaboration with the United States Environmental Protection Agency, the United States Agency for International Development (USAID), and UNEP, has established an Air Quality Management Plan that covers up to 10 districts in Accra and its surroundings [39,41]. Air quality monitoring has been in place in Ghana since 1997, which is the oldest documented system in ECOWAS. This new collaboration has brought the number of monitoring sites up to 16 (although it is claimed that one of them is currently closed). The stations are evenly distributed in order to provide good spatial resolution, and there are plans to strengthen the network even further through a low-cost PM monitoring network distributed across Accra [39]. The current monitoring sites measure air quality in residential areas, on roadsides, and in industrial areas. EPA Ghana expects that reduced mortality rates would follow a reduction in emissions; however, they are aware that the lack of air pollution-related health data is a limitation to this plan. There are plans to close this gap; ID2 mentioned that different ministries in Ghana tend to have a strong, coordinated message when it comes to air quality management.

Ngom et al. [31] explicitly mention the six measurement stations that are currently operating in the city of Dakar, Senegal. These are located in the Guédiaway, Medina, Yoff, Bel-air, HLM, and Cathedral districts, and they are all run by the city’s Air Quality Management Center (CGQA—Centre de Gestion de la Qualité de l’Air). However, Dakar’s CGQA website only mentions five active air quality monitoring sites, excluding the one in Guédiaway [48]. Another study on Dakar combined air quality monitoring data, data from the Ministry of Health, models, and also considered seasonal variability of Saharan dust influence in order to examine patterns of respiratory conditions, such as asthma, bronchitis, and tuberculosis [9]. They found a higher prevalence of respiratory conditions in Dakar versus other parts of the country, which suggested the strong influence of anthropogenic air pollution.

Regarding Nigeria, Aliyu and Botai [47] state that the monitoring network is “scantily distributed”, which implies there are stations, but their location or functionality is not mentioned. Another study [17] shows a figure where two Federal Ministry of the Environment sampling sites are located, in the Rivers State cities of Peter Odili and Abuloma, near Port Harcourt. However, there are no other references to these fixed air quality monitoring stations. Schwela [26] mentions one monitoring station in Lagos that was not operational at that time. In the early 2000s, five stations were installed throughout the country by the Nigerian Meteorological Agency, but it is believed that these have not been functional since 2013, possibly due to insufficient technical know-how that is required to properly maintain equipment [ID5]. In addition, air quality monitoring equipment is sometimes subject to vandalism, which could be a result of the poor public awareness of pollution-related health risks [ID5].

There are plans to improve the situation through the World Bank’s Pollution Management and Environmental Health Plan [49] in order to support Lagos in the establishment of a solid Air Quality Management Plan, starting with the implementation of an air quality monitoring network initially comprised of six fixed sites. The idea is to extend the plan to other Nigerian cities; however, the Air Quality Management Plan will probably comprise a combination of compliance-grade monitoring sites, low-cost sensors, and remote sensing technologies to balance costs [ID5]. Despite these methods not being sufficient on their own, remote sensing has been the main source of air quality surveillance in Nigeria due to the relatively lower price of this technology [ID5].

Countries, such as Burkina Faso and Ivory Coast (especially in the district of Abidjan), mention plans for implementing air quality monitoring, but have yet to realize these goals, or to make information on their development and results publicly available [50,51], [ID3]. The Dynamics-Aerosol-Chemistry-Cloud Interactions in West Africa (DACCIWA) project showed a first effort of combining health and air quality data for Abidjan in spite of lacking official data [ID3]. The project was able to show that exposure to PM_10_ and PM_2.5_ increased the rate of hospital visits, and it was able to predict that decreasing these PM levels to WHO limits could reduce hospital admissions by 4% [50]. Mali recognizes the need for a monitoring network, and it has a program titled “Project of Air Quality Surveillance in the District of Bamako”, led by the National Agency for Sanitation and Pollution Control [26]. There is also an ongoing project led by the Stockholm Environment Institute (SEI) aiming to set up low-cost sensor air quality monitoring sites in Accra (Ghana), Lomé (Togo). and Abidjan (Ivory Coast), the data from which will be open-source and accessible to all. The sensors will be low-maintenance, and the network can easily grow and expand to other countries, allowing for regional capacity building. However, it was noted that low-cost sensors carry with them calibration and accuracy issues that must be monitored closely [ID1], [ID2].

There are other projects and initiatives, such as the Aerosol Robotic Network (AERONET), which has monitoring sites across the world, including in several ECOWAS cities [42,52]. However, the focus is on researching and characterizing aerosols while using aerosol optical depth data that differ from the way health officials use and interpret particulate matter size [52]. Another initiative, the African Monsoon Multidisciplinary Analysis—Couplage de l’Atmosphère Tropicale et du Cycle Hydrologique (AMMA-CATCH) has sites in Mali, Niger, and Benin. Once again, the aims stray from monitoring urban air pollution in relation to health, as the objective is to document long-term climate evolution in West Africa as a whole [53].

In short, actively functioning government air quality monitoring sites seem to be installed in just two of the 15 ECOWAS countries. No information was found for Cabo Verde, Gambia, Guinea, Guinea-Bissau, Liberia, or Sierra Leone. However, there are plans for improving this situation. Amegah and Agyei-Mensah [41] mention initiatives that were launched by the World Bank and the SEI, such as the Clean Air Initiative in Sub-Saharan Africa and the Air Pollution Information Network for Africa, aiming to support nations in this region kickstart their air quality monitoring activities. No specific information was found when investigating these organizations. Plans to improve links to public health surveillance of air pollution-related conditions, such as respiratory diseases, were generally not mentioned.

### 3.3. Regulations

A thorough scrutiny of included articles, government websites, and other gray literature sources revealed the official documents that are summarized in Table 3.

Most ECOWAS countries have general environmental laws published around the 1990s [26]. A number of countries, such as Togo, Benin, and Nigeria, even include the right to a clean environment in their constitution and they have ratified several international environmental agreements, such as the Montreal Protocol on Substances that Deplete the Ozone Layer and the Kyoto Protocol on Climate Change [17,26]. While some nations’ regulations stop at these older generic environmental laws, others have gone significantly further. At the regional level, an ECOWAS-wide pledge to harmonize fuel quality and standards exists as of January 2020, an important step that is expected to be enforced in early 2021 [ID2].

Policies and standards are quite abundant in Nigeria. Action on preserving air quality in general began with the Public Health Act of 1917, along with several pieces of legislation—many of which are still in force today—that followed nation’s independence in 1960 [54]. Environmental legislation continued during the decades that followed, culminating in the creation of the Federal Environmental Protection Agency (FEPA) in 1988. Already in 1991, National Air Quality Standards were promulgated through this institution, but there were no guidelines for PM_2.5_ [45,47]. The Environmental Impact Assessment Act of 1992 was also an important step that would push industries to conduct environmental impact assessments in order to ensure compliance with environmental regulations [17].

The FEPA was replaced by the Federal Ministry of the Environment in 1999. Issues of poor enforcement and general non-compliance with existing regulations had become a concern, as pollution levels at this time were already exceeding both FEPA and WHO regulations [47]. The National Environmental Standards and Regulations Enforcement Agency (NESREA) was established in 2007 in order to strengthen implementation and enforcement in Nigeria, which worked under the Federal Ministry of the Environment with the aim of ensuring that regulation goals were met. Although recognizing a positive step, Ladan [54] highlights that NESREA’s four main environmental priorities do not place air pollution strongly on their agenda. In addition, between 2009 and 2011, 24 national regulations were published by NESREA, including one on bushfires and open burning and another on vehicle emissions, yet only these two pose air quality as a concern in Nigeria [26,55]. New air quality control regulations were published under NESREA in 2014 [56].

Senegal, despite its air quality monitoring efforts in Dakar, has slightly older regulations for air quality. The nation’s Direction de l’Environnement et des Établissements Classés summarizes a list of published standards and regulations [57], and there is also a 2003 standard not included in this list [58].

Ghana’s Air Quality Management Plan, through which the above-mentioned monitoring stations were installed, also includes the publication of National Air Quality Standards and National Motor Vehicle Emissions Standards that are currently undergoing review. Until they are promulgated, air quality control efforts are focused on enforcing existing fuel quality standards and the air quality guidelines that were published in 2000 [39].

Burkina Faso has had active air pollutant emissions standards for automobiles and mopeds since 2001. There are also several ongoing plans aiming to improve urban transportation (and, therefore, urban emissions) and general air quality in the city of Ouagadougou [26,59]. Abidjan (Ivory Coast) is taking on a similar initiative with the World Bank [60]. Ivory Coast and Ghana are also collaborating with the CCAC in order to reduce short-lived climate pollutants and they have published action plans for this [61,62]. Although Ivory Coast does not currently have national standards, WHO guidelines for PM pollutants are generally used as a reference to protect public health [ID3].

Schwela [26] claims that Guinea “at present … has no official criteria for regulating and managing air quality or official emission standards for mobile sources”. In Liberia, hardly any activity regarding urban air pollution regulations has occurred either, yet the government recognizes the urgent need for developing and promoting air quality monitoring [26]. Ivory Coast, Niger, and The Gambia have taken action to reduce the age of their vehicle fleet, restricting the importation of vehicles aged five years and older [41], [ID3]. Benin has air quality standards dating from 2001 that have yet to be promulgated. In Mali, two decrees that were promulgated in 2000 and 2001 address pollution control and management of air pollutants, and fuel standards exist. Yet, no other more precise or more recent information was uncovered other than the fact that the National Agency for Sanitation and Pollution Control has plans to start an action plan to control air quality in Bamako [26]. Little or no information regarding policies, laws, standards, or government action on urban air pollution was found for Cabo Verde, Guinea-Bissau, and Sierra Leone. Table 4, below, shows ambient air quality PM limits for ECOWAS countries with available ambient air quality standards. It can be seen that most countries do not set specific PM_2.5_ standards, and that PM_10_ limits greatly exceed those that were established by the WHO as safe for human health.

### 3.4. Recommendations

Several studies proposed recommendations, solutions, and steps for ECOWAS cities and countries to take in the coming years regarding improving both air quality monitoring capacity and air quality in general.

In an effort to reduce emissions, several studies suggested improving urban mobility through public transport, reducing the age of vehicle fleets, and dissuading the public from using automobiles and motorbikes [41,44,60]. However, this can be difficult to achieve without proper incentive, sound city planning, or economic benefit for the public [ID3]. Efe and Efe [44] and ID5 also suggest addressing other challenges related to urbanization, such as waste management. In some countries, reforming the energy sector is also clearly a matter to be considered. There is a general consensus that establishing clear, strict air quality and emissions guidelines is an important first step. Most studies also agreed that more data—thus, more air quality monitoring sites—is needed in ECOWAS to provide governments with a more accurate picture of their air quality situation. More data would improve the reliability of models and indicators that were used by ECOWAS governments, which could increase stakeholder involvement [ID2].

Increasing available data would also allow for public awareness—and, therefore, public action—to increase, as highlighted by Komolafe et al. [40]. ID5 emphasized that this as an especially important aspect, as some traditional practices contribute to air pollution. These include large-scale crop, land, and forest burning activities, among others. Linking air quality monitoring data to health-related indicators, such as life expectancy—as low as 54 for Nigeria—could also help to call both public and policymaker attention to this issue [ID5]. Proper communication to the general public once these links are established is also key, such as warning citizens when episodes of severe pollution occur so as to protect at-risk groups [9]. Of course, these plans cannot be achieved without establishing long-term funding mechanisms [ID3].

Especially for larger nations, like Nigeria, the need to promote air quality monitoring was strongly highlighted by ID4 and Aliyu and Botai [47]. Amegah [46] advocates for the use of low-cost sensors for inexpensive, high-resolution air quality monitoring for Sub-Saharan Africa in general, and urges UNEP and WHO to push initiatives that promote the use of this technology. Ngom et al. proposed an initial trial of what these new air quality monitoring systems could achieve [31] for Dakar (Senegal), where a hybrid monitoring system, combining the already existing fixed stations with mobile low-cost sensors, was suggested. The Stockholm Environment Institute’s newly funded project, as mentioned above, also sees the potential of low-cost sensors and intends to promote this type of air quality monitoring [ID1].

For countries that have already promulgated standards and regulations, there was also a general call for stricter enforcement and compliance initiatives [39,41,54]. One study suggested introducing a carbon tax in order to sanction non-complying industries and companies [40]. Yakubu [17] suggests the use of external environmental consultants to help ensure compliance amongst different polluting sectors. Gathering information into emissions inventories is also strongly suggested by several studies, as it would allow for better enforcement capacity [28,47], [ID2].

Several participants highlighted the need for promoting both within-country and between-country collaboration. Promoting projects or initiatives that could help to strengthen ties between air quality and health data were also suggested [39], [ID3]. Some countries, like Ivory Coast, Senegal, and Ghana, seem to have set up channels for effective communication amongst their stakeholders. This could be because they each have a government agency that is specifically focused on air pollution management (Centre Ivorien Antipollution for Ivory Coast, CGQA for Senegal, and EPA for Ghana) [ID2,ID3]. However, other countries, like Nigeria, struggle to have all parties on the same page. NESREA, for example, is exempt from enforcing standards in the oil sector, as this industry depends on the Ministry of Petroleum; it seems that communication is lacking or even becomes confrontational between this ministry and the Federal Ministry of the Environment, leading to ineffective strategies [ID5]. The transfer of research to policy action is also questionable in this country [ID4,ID5]; the World Bank’s Pollution Management and Environmental Health Plan is trying to gather ministries and stakeholders in order to create a more synergistic environment in this regard [ID5]. At the regional level, ID3 strongly urged for reinforcing cooperation between ECOWAS countries, lamenting that currently regional communication is not sufficient. ID5 highlighted how other African regions, such as Eastern or Southern Africa, are taking up this regional strategy and are producing strong results, and it suggests for ECOWAS to follow such initiatives.

## 4. Discussion

The above results show that, in general, the status of air quality monitoring, public policy, and legislation in ECOWAS cities is slowly developing, although it has a long way to go before it reaches the levels of Europe, America, Asia, or even other African regions. Research must be promoted, so that the sources and health effects of urban air pollution in ECOWAS can be properly measured and addressed.

As highlighted by numerous studies, the particular geographical location and meteorological conditions of ECOWAS set the region at an initial disadvantage when it comes to air quality [31]. Although less researched than other parts of the world [3], the effects of Saharan dust on the health of ECOWAS citizens cannot be ignored. The seasonal variability of these effects adds to the difficulties that are involved in managing such a situation, even before starting to consider the additional layer of complications introduced by urban anthropogenic air pollution. Aside from the effects on cardiovascular, respiratory, and other aspects of human health, excessive anthropogenic air pollution in a region already struggling with naturally poor air quality has the potential to alter the delicate weather patterns in ECOWAS, which can, in turn, affect agricultural and farming practices, thus also damaging human health in the longer term [27]. The lack of specificity in the information regarding polluting activities of ECOWAS cities, for example, through emissions inventories, makes it challenging to properly understand both the problem at hand and corresponding impact on health.

The results have shown that official air quality monitoring data are lacking or unavailable in most ECOWAS cities, being especially urgent for the larger ones, like Lagos (Nigeria). Some have highlighted how governments in this part of the world may not see investments in air quality monitoring as a priority, given the pressing issues that they face in other policy sectors [46]. Infectious diseases, widespread poverty, and social deprivation, indeed, force governments to have other matters constantly on the top of their agendas. In the case of Liberia, for example, the civil war ending in 1996, along with ongoing conflicts, result in an unstable political situation, in which, understandably, air quality management cannot be set as a priority. It must also be added that not having air quality data in the first place also makes it easier for the issue of air pollution to go politically unnoticed. Moreover, when it comes to studying air pollution and related health effects, Amegah [41] also claims that Sub-Saharan Africa—and the present study has shown that ECOWAS is particularly vulnerable to this—is systematically excluded from such research, setting the region further behind. Not having air quality or health data to highlight the severity of the situation also makes creating public awareness difficult. There are traditional practices that contribute immensely to both household and ambient air pollution in ECOWAS countries; social change in this matter requires up-to-date information to be effectively communicated to the public and their representatives, along with government-backed incentives and funds that are designed to help the adoption of new habits. Structural investments in these urban settings, for instance, in waste management and energy provision, are also a key dimension of any comprehensive policy.

The prohibitive costs of the infrastructure necessary to establish air quality monitoring networks also justifiably dissuade governments from investing in them [28]. Once set up, there is also a long list of associated requirements, from energy needs to technical maintenance and human capacity, all of which must be strengthened in ECOWAS in order to ensure that air quality monitoring equipment continue to function. Solutions, such as low-cost sensors, are slowly spreading across ECOWAS cities. Although they appear to be a promising option to speed up the development of air quality monitoring networks, these devices carry several issues that must not be overlooked. Calibration issues can lead to unreliability, especially if there is not a reference monitor to measure against. Moreover, standards for these devices have yet to be developed, which can lead to the coexistence of sensors of varying quality [46]. These sensors can, at least, serve as a starting point for air quality monitoring in ECOWAS, with the goal of transitioning into more permanent monitoring networks. What is clear is that increasing the availability of air quality data would be extremely useful for ECOWAS governments to improve the quality of their models, strategize on mitigation measures, target and prioritize among emissions sources, and increase stakeholder involvement. Finding an adequate combination of the different air quality monitoring equipment available to date, from research-grade equipment, to low-cost sensors, to satellite imagery, to models, could represent a cost-effective compromise for ECOWAS countries. Further research is needed in order to determine if such an amalgam has the potential to be feasible, affordable, and of acceptable quality.

It has also been highlighted that, even if more ambient air quality data are acquired through low-cost sensors or through other means, simply gathering air quality data would not be sufficient for triggering effective action against air pollution. Firstly, specific data from multiple sectors that influence air quality is necessary, such as agriculture, industry, and transportation. Coordination with the health sector should also be promoted to further the use of the data, as proposed by the DACCIWA project or by a study conducted in Senegal [9,27]. Neither the availability of health data nor the level of coordination between different ministries whose actions may influence health were mentioned in most studies; they were even highlighted as a concerning gap by some. Moreover, even if the data were available, there must be long-term funding mechanisms in place in order to ensure that it can be properly used and exploited. The process of acquiring data within each of these sectors will undoubtedly encounter numerous additional barriers along the way. A potential opportunity lies within the fact that most ECOWAS countries do not have air quality monitoring or detailed health information systems in place yet: they can, therefore, be designed in a coordinated fashion from the beginning, rather than made later out of a merge of two fragmented systems. Although a positive outlook, challenges that are related to the establishment of such systems within such a complex institutional landscape will undoubtedly arise.

Additionally, since data alone do not make a policy, proper attention and broad vision by policymakers and administrations is also paramount in ensuring successful urban air quality management, as improving legislation is a necessary first step. Some ECOWAS nations include the right to a clean environment in their very constitutions, and others have air quality regulations that date back to the 1980s and 1990s. Despite this strong start, these regulations are now outdated, with, for instance, PM limits that are greatly above WHO guidelines. Therefore, even in countries with published standards, public health remains insufficiently protected. The question of health literacy arises due to this situation, as what is officially considered to be acceptable air quality for these nations is already far beyond the threshold of what health guidelines consider to be so. This shows that the transfer of research and evidence into policy making encounters particular difficulties and delays in ECOWAS countries.

Moreover, there are many countries that have yet to publish air quality standards at all. Issues of fragile governance and weak enforcement capabilities must be added to these limited regulations: even when legislation is present, ensuring compliance is often a challenge due to the limited administrative resources. This is, in part, due to the fact that urban air pollution is a matter that affects an extremely wide range of sectors. Therefore, its management requires a high level of coordination and cooperation amongst them. Having an agency that is responsible for centralizing air quality management, like in the cases of Senegal, Ghana, and Ivory Coast, could lead to positive results in this respect, although detailed evaluation studies are lacking. City-level regulations or initiatives, only mentioned for Abidjan, could also serve as an interesting means of improving governance, as efforts at the local level could be less fragmented and, therefore, more effective. However, even when air pollution management is relatively centralized like in Nigeria, other aspects, like country size, the importance of industrial activity (backed by influential private networks and lobbies that are difficult to regulate), and low public awareness make enforcing regulations just as difficult. The added challenge of managing the very strategic oil sector puts the health of the citizens of the most urbanized country in ECOWAS in a delicate position. It is clear that not only a strong science-policy interface is needed—lacking in ECOWAS countries according to several experts—but also political will and strong means are essential in making progress on air quality management.

Interestingly, different studies view the problems of regulating air pollution through different lenses. While Ladan [54] clearly sees sectorial enforcement issues to be paramount, others view the problem as a result of poor urban planning [44], while others highlight the effects on the environment and climate change [40]. Although general health effects are mentioned in most studies, the source of the problem—and the corresponding path towards the solution—varies greatly according to different authors, and the focus often strays from public health. A clear example is Nigeria’s shift from the Public Health Act addressing air quality to NESREA’s dependence on the Federal Ministry of the Environment. Failing to view air pollution as a public health problem in and of itself, rather than as a consequence of a different problem, threatens to remove attention from its life-threatening effects. However, the multiple ways in which it can be framed can also be seen as an advantage, since it can be tailored to different audiences and stakeholders, with the end result of improving public health regardless of the initial motivation. The growing global movement to mitigate climate change, for example, has the potential to advance the air pollution agenda, since traffic, energy, and industrial emissions are generally damaging to both causes [40,65]. However, the immense threat to public health may not be neglected, and research into the immediate and long-term health effects of urban air pollution in ECOWAS is highly needed in any case.

This scoping study has identified the need for further research into the status of urban air quality management in ECOWAS. Recommendations that were provided by literature and experts, which can serve to guide future research, have also been highlighted. These include promoting collaboration at various levels. While encouraging multi-stakeholder implications within ECOWAS is an ideal strategy, cooperation initiatives have generally occurred between the North and the South, such as USAID’s Air Quality Management Plan in Accra (Ghana) or the World Bank’s Pollution Management and Environmental Health Plan in Lagos (Nigeria). An unfortunate side of this is to observe the disparity that exists within ECOWAS countries, as some benefit from international partnerships while others struggle to lift their projects off the ground. The encouraging progress of countries that are involved in these programs may send the message that only those receiving this type of aid are able to advance in their air quality control plans. Placing efforts into promoting regional cooperation, or into programs that can benefit multiple nations at once, could help ECOWAS as a whole create more sustainable improvements. It should also be remembered that the natural pollution in the region, as well as the pollution resulting from desertification, makes transboundary pollution effects extremely relevant; therefore, the progress a certain country makes can be limited to what is ongoing in neighboring ones. This can serve as additional motivation to share resources and combine efforts when it comes to addressing air pollution. Furthermore, reviewing the literature findings, it seems that countries with laws or regulations on air quality have not yet implemented air quality monitoring, while countries with air quality monitoring stations have yet to promulgate their standards. In the end, it could seem difficult to take effective action in either situation: for countries with regulations, it is difficult to know whether they are being met without proper air quality monitoring and, for those countries that do monitor air quality, there are no standards to compare the acquired data to. This paradoxical situation could actually present as an additional opportunity for cooperation amongst ECOWAS nations, where the strengths of the first group of countries can serve to address the gaps of the second, and the other way around. Despite these findings, it is clear that further research, including local insights from experts in the field, is needed in order to better understand upcoming steps that ECOWAS countries should take when it comes to urban ambient air quality monitoring.

Finally, ECOWAS could draw from the example of other African countries—and even entire regions—that can serve as a roadmap for their combined air pollution-combatting efforts. South Africa, for example, has a well-distributed air quality monitoring network, which, as mentioned before, has led to studies that use official air quality monitoring data in order to assess air quality and corresponding health effects [29,41]. Several other African nations also have air quality monitoring stations in place: Botswana began monitoring air quality in 1996 and, today, has 21 stations, and both Ethiopia and Tanzania have eleven installed air quality monitoring stations. In addition, these countries use WHO or United States EPA guidelines until their own air quality standards are promulgated [26]; it should be noted, however, that data on ambient air pollution-related health issues in these countries is lacking in most cases, as public health surveillance efforts continue to focus on infectious diseases or indoor air pollution [66,67,68,69,70]. At the regional level, it has been mentioned how Northern, Eastern, and Southern African countries have strived for a regional approach in order to address urban air pollution. While ECOWAS has come together on economic and financial matters [71], such collaboration on this front is only slowly taking off. Recent initiatives, such as the West and Central Africa Regional Framework Agreement on Air Pollution, signed in Abidjan in 2009, showed promise for a regional approach to addressing urban air pollution [72]. The ECOWAS-wide pledge to reduce sulfur in fossil fuels [73], which was signed this year, was also a strong statement. However, it is unsure whether substantial concrete action has occurred. Stronger cooperation, policies adapted to the local context, and strategies informed by precise data and knowledge are crucial in helping ECOWAS, and Africa as a whole, fight urban air pollution and its health effects in a resolute and effective way.

### Limitations

The main limitation of this study refers to the low level of information available regarding the status of urban air quality monitoring, policy, and health effects across ECOWAS countries. Most states have little or no available literature that met the inclusion criteria, as most included literature covered larger nations, like Nigeria. Although this highlights the need for further research, it makes it difficult to determine each country’s specific situation. Out-of-date government websites, lacking access to official legislation, and insufficient local insights added to the difficulties in obtaining reliable information on the status of air pollution in ECOWAS cities.

Additionally, the aim and scope of the study could have led to the exclusion of studies that more directly addressed the health effects of urban air pollution in ECOWAS. However, including these would have strayed from the objective of assessing the status of official air quality monitoring and policy in ECOWAS cities. Likewise, studies focusing exclusively on indoor air pollution may have provided some insight into the ambient air quality of ECOWAS cities or on health effects, yet including them would have strayed too far from the aims and scope of the present review.

## 5. Conclusions

Urban air pollution is a global health issue that causes millions of preventable deaths worldwide. The status of air quality monitoring sites, standards, and polluting activities, in relation to health in ECOWAS cities, has been thoroughly examined through this scoping study. The overall limited amount of data, legislation, and country-specific polluting activities make it difficult for this region to be fully aware of the health threats that their population faces. Simply providing these countries with air quality data is not sufficient: it is also vital to promote sustainable initiatives that can increase data availability in the long term, strong communication between different government sectors and stakeholders, and ECOWAS-wide collaboration. This way, government action, adequate legislation, and public awareness can improve, which leads to the region as a whole prioritizing the protection of urban air quality and public health.

## Figures and Tables

**Figure 1 ijerph-17-09151-f001:**
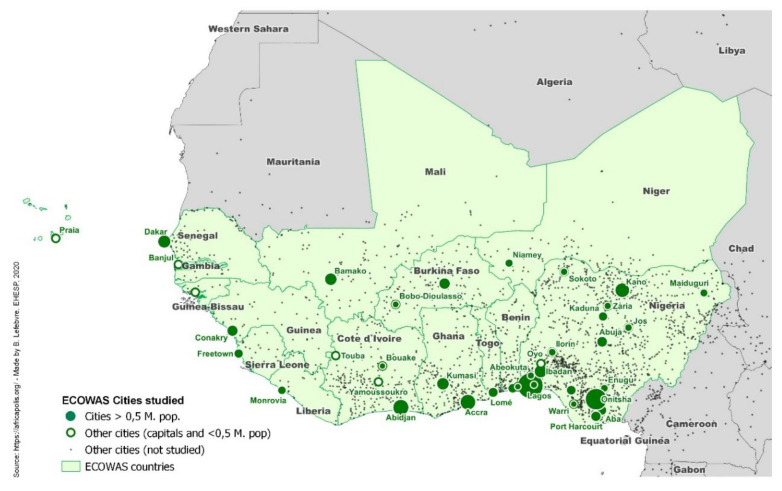
Economic Community of West African States (ECOWAS) urban system and cities selected for the study.

**Figure 2 ijerph-17-09151-f002:**
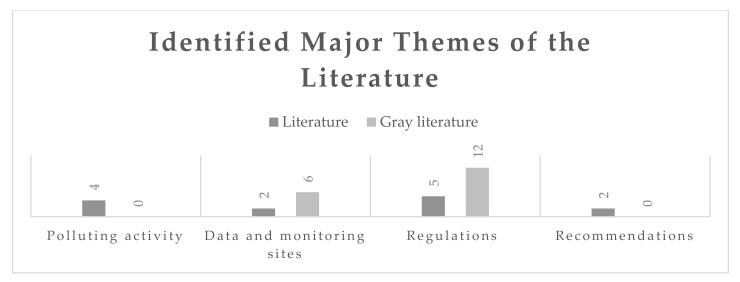
Main themes of the scoping review as found in the literature.

**Table 1 ijerph-17-09151-t001:** Population growth rate and urbanization and industrialization levels of ECOWAS countries.

ECOWAS Country	Population (Million) ^1^	% Population Change in 2020 ^1^	% of Population in Urban Areas ^2^	Industry as % of GDP * ^3^
Benin	12.12	2.73%	49	13
Burkina Faso	20.90	2.86%	29	23
Cabo Verde	0.56	1.10%	50	18.8
Ivory Coast	26.38	2.57%	49	21.3
The Gambia	2.42	2.94%	56	12
Ghana	30.07	2.15%	52	29
Guinea	13.13	2.83%	37	46.5
Guinea-Bissau	1.97	2.45%	34	13.5
Liberia	5.06	2.44%	42	5.4
Mali	20.25	3.02%	32	24
Niger	24.21	3.84%	17	14.2
Nigeria	206.14	2.58%	52	43
Senegal	16.74	2.75%	51	22.7
Sierra Leone	7.98	2.10%	37	18.6
Togo	8.28	2.43%	50	33.7

^1^ From [23]. ^2^ From [24]. ^3^ From [25]. * Gross Domestic Product.

**Table 2 ijerph-17-09151-t002:** Identification numbers assigned to expert participants.

ID * Number	Country/Region of Expertise
ID1	Ghana, Ivory Coast, Togo
ID2	Ghana, Ivory Coast, Togo
ID3	Ivory Coast
ID4	Nigeria
ID5	Nigeria

* Identification.

**Table 3 ijerph-17-09151-t003:** Summary of studies and regulations included for review, per country and region.

Country	Scientific Literature	Gray Literature	Monitoring Sites Identified	Polluting Activity Identified	Identified Air Quality Standards	Discussed in Interviews
Benin	-	1	-	Yes	2001 Air Quality Standards	Work has been done to develop emissions inventories ^1^
Burkina Faso	-	3	-	Yes	2001 Vehicle Emission Standards	-
Cabo Verde	-	-	-	-	-	-
Ivory Coast	-	3	-	Yes	-	Strength of the Centre Ivorien Antipollution ^1,2^. DACCIWA project a successful linking air quality and health. Similar research ongoing ^2^
Gambia	-	-	-	-	-	-
Ghana	-	2	Yes	Yes	National Air Quality Standards under review	Strong coordination amongst government actors ^1^
Guinea	-	-	-	Yes	-	-
Guinea-Bissau	-	-	-	-	-	-
Liberia	-	-	-	-	-	Work has been done to develop emissions inventories ^1^
Mali	-	-	-	Yes	2001 decree addresses air pollution management	-
Niger	-	-	-	-	-	-
Nigeria	9	3	-	Yes	2014 Air quality control regulations	Difficult challenges and barriers ^3^. Recent plans and progress ^4^
					2011 Vehicle emissions regulations	
Senegal	2	3	Yes	Yes	2003 Norme Senegalaise	-
Sierra Leone	-	-	-	-	-	-
Togo	-	-	-	Yes	-	Plans for a low-cost sensor network ^5^
Sub-Saharan Africa	2	3	-	-	-	-
ECOWAS	-	-	-	Yes	-	Harmonization of fuel standards discussed ^1^

^1^ [ID2]; ^2^ [ID3]; ^3^ [ID5]; ^4^ [ID1]; ^5^ [ID4]. * Dynamics-Aerosol-Chemistry-Cloud Interactions in West Africa.

**Table 4 ijerph-17-09151-t004:** Standards for health and air pollution comparing ECOWAS and World Health Organization (WHO).

Country/Agency	PM *_2.5_ (µg/m^3^)		PM_10_ (µg/m^3^)	
	Annual Mean	24-h Mean	Annual Mean	24-h Mean
WHO	10	25	25	50
Benin ^1^	-	-	50	230
Burkina Faso ^2^	200–300 ^3^			
Nigeria ^4^	-	-	60	150
Senegal ^5^	-	-	80	260

^1^ From [63]. ^2^ From [64]. ^3^ The word “particulates” was simply mentioned, without specifying size or exposure time. ^4^ From [56]. ^5^ From [58]. * Particulate Matter.

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
