# Peer review of "A Scoping Review on Air Quality Monitoring, Policy and Health in West African Cities"

_ijerph, 2020, doi:10.3390/ijerph17239151_

Round 1

Reviewer 1 Report

This is an outstanding, thorough, comprehensive scoping review covering a very important gap in the research literature.  I have no suggestions for changes, other than minor editing for style.

Author Response

Reviewer comment: This is an outstanding, thorough, comprehensive scoping review covering a very important gap in the research literature.  I have no suggestions for changes, other than minor editing for style.

Author answer: Thank you very much for this positive answer. We are delighted that you linked our manuscript.

Reviewer 2 Report

A well written and useful review paper for those doing work in West African Pollution studies.

line 35: anyone who doesn't know what PM will also not know that PM2.5 and PM10 refers to particles with diameters less than 2.5 or 10 microns respectively. This should be added.

Lines 132-136: this is a matter of taste, but I think these sub questions are best presented as a bulleted list rather than sentences in a paragraph.

Line 193: The fact that 1076 articles were found is completely irrelevant if most were duplicates. Just state the number found AFTER duplicates were removed.

Line 195: what is meant by "titles were screened". This seemed to have eliminated many papers, so the nature of this screening is important.

Line 314: I think "limit" should be replaced with "reduce"

Line 325: I think the statement that AERONET data is not focussed urban pollution and health is a little misleading: AERONET data actually has nearly all of the information needed, it's just not packaged in a way that health officials are used to seeing. It provides a continuous distribution of diameters rather than a focus on the mass with a diameter below 2.5 microns, for example. When available the data IS HIGHLY usable for application to urban health, it just needs to be translated to a vocabulary the health community is familiar with. The larger issue is that the data is not available under cloudy skies, and even with partially cloudy skies the data is limited to aerosol optical depth only, which not as useful to the health community as size parameters.

Line 446: "chanels" > "channels"

Table A1: there seems to be an unlabelled section between "free text terms" and "controlled vocabulary terms"

Table A2: does each paper really fit under only one category? Seems unlikely.

Author Response

Dear Reviewer,

thank you very much for your very useful suggestions to further improve our manuscript. Kindly find below point by point our response to your suggested changes.

Reviewer comment: A well written and useful review paper for those doing work in West African Pollution studies. Line 35: anyone who doesn't know what PM will also not know that PM2.5 and PM10 refers to particles with diameters less than 2.5 or 10 microns respectively. This should be added.

Authors answer: We have added a description in line 35-43.

Reviewer comment: Lines 132-136: this is a matter of taste, but I think these sub questions are best presented as a bulleted list rather than sentences in a paragraph.

Authors answer: We agree to your style suggestion and changed it accordingly in line 142-146

Reviewer comment: Line 193: The fact that 1076 articles were found is completely irrelevant if most were duplicates. Just state the number found AFTER duplicates were removed.

Authors answer: We changed this according to your suggestion 

Reviewer comment: Line 195: what is meant by "titles were screened". This seemed to have eliminated many papers, so the nature of this screening is important.

Authors answer: We removed it and changed the text accordingly in line 204-208

Reviewer comment: Line 314: I think "limit" should be replaced with "reduce"

Authors answer: We have changed it in line 331

Reviewer comment: Line 325: I think the statement that AERONET data is not focussed urban pollution and health is a little misleading: AERONET data actually has nearly all of the information needed, it's just not packaged in a way that health officials are used to seeing. It provides a continuous distribution of diameters rather than a focus on the mass with a diameter below 2.5 microns, for example. When available the data IS HIGHLY usable for application to urban health, it just needs to be translated to a vocabulary the health community is familiar with. The larger issue is that the data is not available under cloudy skies, and even with partially cloudy skies the data is limited to aerosol optical depth only, which not as useful to the health community as size parameters.

Authors answer: We have added to the argument in line 342-243

Reviewer comment: Line 446: "chanels" > "channels"

Authors answer: Thanks for noticing it is changed in line 468

Reviewer comment: Table A1: there seems to be an unlabelled section between "free text terms" and "controlled vocabulary terms"

Authors answer: There was an unnecessary division in the table that has been removed.

Reviewer comment: Table A2: does each paper really fit under only one category? Seems unlikely.

Authors answer: In the methods section (lines 168-172), it is stated that if a study touched upon more than one category the most prominent one was selected for categorization.

Reviewer 3 Report

The authors presented a review using a rigorous methodology and screening a huge amount of documents.
I think that the manuscript is clear and well written. The scope of the review isclearly stated and, in may opinion, it has been achieved.

I have two suggestions related to the framework description for this review, reported in the introduction:

- line 39: in the previous sentences the authors focused on PM pollution issue, thus I suggest to change reference [5], which regards NOx, with one more appropriate. I t can be foun, e.g., using the following searching criteria on Scopus: particulate matter+Covid 19+health (in TITLE-ABS-KEY). Moreover I suggest a brief separated discussion about gaseous pollutants&health: e.g. NOx, SOx, O3.

- Using the following searching criteria on Scopus: air quality+Africa(in TITLE-ABS-KEY) in review only, several matches can be found. Please cite one or more if appropriate for comparison to the present review.

Author Response

Dear Reviewer,

thank you very much for your very useful suggestions to further improve our manuscript. Kindly find below point by point our response to your suggested changes.

The authors presented a review using a rigorous methodology and screening a huge amount of documents. I think that the manuscript is clear and well written. The scope of the review is clearly stated and, in my opinion, it has been achieved. I have two suggestions related to the framework description for this review, reported in the introduction:

Reviewer comment: - line 39: in the previous sentences the authors focused on PM pollution issue, thus I suggest to change reference [5], which regards NOx, with one more appropriate. I t can be foun, e.g., using the following searching criteria on Scopus: particulate matter+Covid 19+health (in TITLE-ABS-KEY). Moreover I suggest a brief separated discussion about gaseous pollutants&health: e.g. NOx, SOx, O3.

Authors answer: We have added a section in line 40-43 to cover your suggestion.

Reviewer comment: - Using the following searching criteria on Scopus: air quality+Africa(in TITLE-ABS-KEY) in review only, several matches can be found. Please cite one or more if appropriate for comparison to the present review.

Authors answer: Thanks for pointing that out we have added the examples and addition in line 118-121